# Prevalence, Risk Factors, and Pathophysiology of Nonalcoholic Fatty Liver Disease (NAFLD) in Women with Polycystic Ovary Syndrome (PCOS)

**DOI:** 10.3390/biomedicines10010131

**Published:** 2022-01-07

**Authors:** Svetlana Spremović Rađenović, Miljan Pupovac, Mladen Andjić, Jovan Bila, Svetlana Srećković, Aleksandra Gudović, Biljana Dragaš, Nebojša Radunović

**Affiliations:** 1Clinic for Gynecology and Obstetrics, University Clinical Centre of Serbia, 11000 Belgrade, Serbia; spremovics@gmail.com (S.S.R.); andjicmladen94@gmail.com (M.A.); Bilamsj@gmail.com (J.B.); sasagudovic@gmail.com (A.G.); 2Department of Gynecology and Obstetrics, Faculty of Medicine, University of Belgrade, 11000 Belgrade, Serbia; 3Center for Anesthesiology and Resuscitation, University Clinical Centre of Serbia, 11000 Belgrade, Serbia; svetlanasreckovic@yahoo.com; 4Intermedicus BIS, Specialized Hospital for Infertility, 11000 Belgrade, Serbia; biljatv@gmail.com (B.D.); radunn01@gmail.com (N.R.); 5Serbian Academy of Science and Art, 11000 Belgrade, Serbia

**Keywords:** PCOS, NAFLD, obesity, insulin resistance, hyperandrogenemia

## Abstract

**Background**: Polycystic Ovary Syndrome (PCOS) is one of the most common endocrine disorders in women’s reproductive period of life. The presence of nonalcoholic fatty liver disease NAFLD, one of the leading causes of chronic liver disease in the Western world, is increased in women with PCOS. This review aims to present current knowledge in epidemiology, pathophysiology, diagnostics, and treatment of NAFLD in PCOS with an emphasis on the molecular basis of development of NAFLD in PCOS women. **Methods**: Authors investigated the available data on PCOS and NAFLD by a MEDLINE and Pub Med search during the years 1990–2021 using a combination of keywords such as “PCOS”, “NAFLD”, “steatohepatitis”, “insulin resistance”, “hyperandrogenaemia”, “inflammation”, “adipose tissue”, and “obesity”. Peer-reviewed articles regarding NAFLD and PCOS were included in this manuscript. Additional articles were identified from the references of relevant papers. **Results**: PCOS and NAFLD are multifactorial diseases, The development of NAFLD in PCOS women is linked to insulin resistance, hyperandrogenemia, obesity, adipose tissue dysfunction, and inflammation. There is the possible role of the gut microbiome, mitochondrial dysfunction, and endocannabinoid system in the maintenance of NAFLD in PCOS women. **Conclusions**: There is a need for further investigation about the mechanism of the development of NAFLD in PCOS women. New data about the molecular basis of development of NAFLD in PCOS integrated with epidemiological and clinical information could influence the evolution of new diagnostic and therapeutic approaches of NAFLD in PCOS.

## 1. Introduction

Polycystic Ovary Syndrome (PCOS) is one of the most common endocrine disorders in women of childbearing age [1]. It has been estimated that around 6–10% of women in the reproductive period are affected by this endocrinological disease considering the classical definition of the syndrome and the prevalence rise to 18–20% when used the Rotterdam criteria [2,3,4]. The women with PCOS are characterized by ovulatory dysfunction, hyperandrogenism, and/or polycystic ovary morphology confirmed by ultrasonography [5]. Hyperandrogenism usually manifests as hirsutism, acne, and/or alopecia, whereas ovulatory dysfunction is expressed as oligomenorrhea or amenorrhea and subfertility [6,7]. Besides the reproductive manifestation of PCOS, obesity and insulin resistance are present in this syndrome; it has been considered that both have main roles in the pathogenesis of PCOS [6,8,9]. Recent findings suggest that PCOS is a multi-system reproductive metabolic disease that includes other health issues such as cardiovascular disease, diabetes mellitus, and cerebrovascular disease [10,11,12,13].

Another disorder associated with insulin resistance is nonalcoholic fatty liver disease (NAFLD) [14], one of the most common causes of chronic liver disease in the Western world with a prevalence of around 6.3–33% in the general population [15,16]. This disease represents a spectrum of disorders, which include not just benign forms such as hepatic steatosis (fat accumulation in liver tissue without inflammation) but also steatohepatitis (fat accumulation in liver tissue with inflammation and hepatocellular injury) with or without fibrosis, which could lead to liver cirrhosis and possibly hepatocellular carcinoma [17,18]. Although NAFLD is a rising health issue, the true pathogenesis of NAFLD is not clear, but it has been indicated that unhealthy lifestyle, obesity, dyslipidemia, and ethnicity are risk factors for NAFLD development, and similar to PCOS, insulin resistance has a pivotal role in NAFLD’s pathogenesis [14,17,18]. Interestingly, NAFLD has associations with extrahepatic manifestations and endocrinopathies such as hypothyroidism, hypopituitarism, growth hormone deficiency, hypercortisolism, and primarily with PCOS [19]. Similar to PCOS, NAFLD is strongly associated with obesity and insulin resistance as well as cardiovascular disease and diabetes mellitus type 2 [17,20,21]. The presence of NAFLD is increased in women with PCOS, especially in women with high serum androgen levels, obesity, and insulin resistance; thus, it is hypothesized that these multifactorial conditions have one or more contact points rather than a full coexistence [4].

Therapeutic limitations due to the presence of NAFLD have multiple clinical consequences for the patient with PCOS, which doctors face through clinical work. The most commonly used therapy for irregular uterine bleeding or amenorrhea in patients with PCOS is a combination of oral contraceptives with antiandrogenic action, which can be used for a long time as they do not worsen metabolic parameters [22]. High ALT, which is sometimes a consequence of NAFLD, is a contraindication for the use of contraceptives, unfortunately in a situation that requires urgent and effective therapy. Normalization of BMI, insulin resistance, and androgens, therapy with probiotics, and a correction of laboratory parameters resulting in improvements of both PCOS and NAFLD.

Taking into account how great the health issues of NAFLD and PCOS are and their increased prevalence worldwide, the aim of this review is to present the epidemiology, pathophysiology, diagnosis, and treatment of NAFLD in PCOS concerning the possible molecular basis of development of NAFLD in PCOS women.

## 2. Methodology

The authors investigated the available data on PCOS and NAFLD by a MEDLINE and Pub Med search during the years 1990–2021 using a combination of keywords, such as “PCOS”, “NAFLD”, “steatohepatitis”, “insulin resistance”, “hyperandrogenaemia”, “inflammation”, “adipose tissue”, and “obesity”. Randomized controlled studies were used when available; otherwise, the most relevant literature on this topic was included based on the authors’ evaluations. Peer-reviewed articles regarding NAFLD and PCOS were included in this manuscript. Additional articles were identified from the references of relevant papers. 

## 3. Epidemiology and Predictors of NAFLD in PCOS Women

The increased prevalence, as well as the health importance of NAFLD and PCOS, make both diseases prominent in current epidemiological and clinical investigations.

The first mention of a possible link between NAFLD and PCOS was the case report of a biopsy diagnosing nonalcoholic steatohepatitis in a woman with PCOS and insulin resistance [23]. The patient underwent a diet and exercise program which led to weight loss and normalization of aminotransferase levels. The biopsy of the liver was repeated after 13 months and revealed a reduction of steatohepatitis and inflammation. Taking into account the same risk factors and interplay between NAFLD and PCOS, numerous studies have investigated the prevalence, risk factors, characteristics, and predictors of NAFLD in PCOS women.

Won et al. [24] settled a retrospective cohort study aimed at evaluating the prevalence and predictors of NAFLD in PCOS women. The study included 586 women with PCOS, which documented the PCOS phenotype, metabolic syndrome (MetS) diagnosis, body composition, insulin sensitivity, sex hormones, lipid profile, liver function, and transient elastography (TE) in the cohort. They have observed that MetS diagnosis and hyperandrogenism were the risk factors associated with NAFLD occurrence, whereas insulin levels (assessed through the 75 g glucose tolerance test) and body mass index were not significant risk factors. The same authors observed that higher aspartate aminotransferase was not presented until late liver damage.

Romanowski et al. [25] performed a case-control study to determine the prevalence of NAFLD and metabolic syndrome and the association of NAFLD with metabolic syndrome components in NAFLD disease. The case group consists of 101 PCOS women while the control group consists of 30 non-PCOS, healthy women. The case group was divided into two groups PCOS + NAFLD and PCOS alone. The prevalence of NAFLD was significantly higher in PCOS women, but there was no difference in the prevalence of metabolic syndrome in women with and without PCOS. PCOS women with NAFLD have higher age, weight, BMI, abdominal circumference, insulin levels, and glucose intolerance than PCOS women without NAFLD. Metabolic syndrome was associated with NAFLD in PCOS women.

Asfari et al. [26] evaluated the association between NAFLD and PCOS using a large national database. The study included 50,785,354 women, from which 7415 (0.15%) had PCOS. Women with PCOS were younger and more obese than non-PCOS women. The rate of NAFLD was significantly higher in PCOS women.

Shengir et al. [27] investigated the prevalence and predictors of NAFLD and fibrosis in women with PCOS from South Asia, a population with severe NAFLD and PCOS. The screening program for NAFLD and fibrosis in 101 PCOS women was conducted by transient elastography with an associated controlled attenuation parameter (CAP). NAFLD was defined as CAP ≥ 288 decibels per meter. Significant liver fibrosis (stage 2 and higher out of 4) was defined as TE measurement ≥ 8.0 kilopascals. The prevalence of NAFLD and liver fibrosis in this study group were 39.6% and 6.9%, respectively. After multiple regression and adjusting for duration of PCOS and insulin resistance measured by a homeostasis model for assessment of insulin resistance, the independent predictors for development of NAFLD in South Asian women with PCOS were higher body mass index, hyperandrogenism, and elevated ALT.

Salva Pastor et al. [28] settled a cross-sectional study aimed at determining the frequency of NAFLD in the Mexican population with PCOS and matched controls. This cross-sectional study included 98 women in the age range of 18–44 years old. The exclusion criteria were significant alcohol consumption, chronic liver disease, use of steatogenic drugs, and PCOS treatment or fertility protocol. The control group consisted of women matched by age and body mass index. The transient elastography was used for NAFLD diagnosis. Similar to results reported from other studies, the NAFLD prevalence and the risk for NAFLD were significantly higher in the PCOS group. The hyperandrogenism was higher in PCOS women. The prevalence of NAFLD was significantly higher in phenotype A phenotype of PCOS which include hyperandrogenism, ovulatory dysfunction, and polycystic ovaries, (about 84.3%) while the prevalence of NAFLD in other phenotypes was about 41.1%.

Considering that there is a lack of data about the prevalence of PCOS in overweight and obese premenopausal women with NAFLD, Vassilatou et al. [29] conducted prospective, observational, and cross-sectional studies aimed at the evaluated prevalence of PCOS in this group of women. The 110 women were studied for the presence of NAFLD and PCOS and about 64.5% of women had NAFLD. The prevalence of PCOS in women with NAFLD was higher than in women without NAFLD.

Harsha Varma et al. [30] found that the prevalence of NAFLD is about 38.3% in PCOS women. They reported that serum total cholesterol, insulin, HOMA-IR, hyperandrogenism, ALT, and AST are factors associated with NAFLD, but after multiple regression analyses, only HOMA-IR and hyperandrogenism were significant predictors of NAFLD. Sarkar et al. [31] reported that PCOS is associated with severe nonalcoholic steatohepatitis (NASH) including fibrosis, suggesting that PCOS women should be screened for NAFLD.

Macut et al. [32] evaluated the risk factors associated with NAFLD in Caucasian women. They settled the cross-sectional study with 600 PCOS women who underwent anthropometric measurements and analyses of insulin, fasting blood glucose, total testosterone, SHBG, liver function, lipid accumulation product, and insulin resistance markers. The prevalence of NAFLD was higher in women with PCOS than in controls (50.6% versus 34.0%, respectively). The waist circumference, insulin, HOMA-IR, total cholesterol, and triglycerides were higher than in the control group. The NAFLD correlated significantly with waist circumference, body mass index, glucose, HOMA-IR, triglycerides, and lipid accumulation product in PCOS women.

Rocha et al. [4] conducted a meta-analysis aimed at exploring which characteristics of PCOS, androgen excess, metabolic derangements, or both influence NAFLD risk. They have included 17 studies with 2734 PCOS patients and 2561 controls of similar age and body mass index. It has been observed the increased prevalence of NAFLD in PCOS patients (odds ratio 2.54, 95% confidence interval 2.19–2.95). The highest prevalence of NAFLD was among PCOS women with hyperandrogenemia. PCOS women with NAFLD have higher serum total testosterone and free androgen index than PCOS women without NAFLD. The studies which used multivariate analysis observed that serum androgens are independent predictors of NAFLD in PCOS women.

Shengir et al. [33] evaluated the association between PCOS and NAFLD in premenopausal women. This meta-analysis included 23 studies with 7148 participants. The results obtained in this meta-analysis showed that PCOS women had a 2.5-fold risk of NAFLD compared to controls. The South American/Middle East PCOS women had a greater risk of NAFLD than women of European and Asian origin. Body mass index seems to be the main cofactor for the development of NAFLD in premenopausal PCOS.

Wu et al. [34] settled the meta-analysis aimed at investigating whether NAFLD is associated with PCOS as a consequence of shared risk factors or whether PCOS contributes to NAFLD as an independent factor. The meta-analysis included 17 studies. The risk for NAFLD was higher in women with PCOS. After stratifying by body mass index and geographic origin of the studied population, the risk for NAFLD was significantly higher in the obese subjects, nonobese subjects, subsets from Europe and the Asia-Pacific region, and patients from America. Higher risk was observed for NAFLD only in hyperandrogenism patients with PCOS. The studies on epidemiology and risk factors for development of NAFLD in PCOS women are presented in Table 1.

## 4. Pathophysiology of NAFLD in PCOS

Insulin resistance has the main role in the development of NAFLD and PCOS and is the present link between these two pathological entities [35]. It has been estimated that about 80% of NAFLD patients have insulin resistance and that insulin levels are the independent factor for the presence of NAFLD [36,37]. Insulin resistance leads to hyperinsulinemia which on the other hand leads to a decrease in mitochondrial fatty acid oxidation, generation of inflammation, necrosis, and fibrosis that ultimately leads to the progression of NAFLD [38]. This suggests that the development of NAFLD is the consequence of the activation of multiple pathways which include inflammation, oxidative stress, adipocytokine, etc. [39]. The disturbance in insulin signaling is not located only in the liver, it also covers muscle. Insulin signaling in the liver is impaired by lipotoxic accumulation [40,41]. Decreased mitochondrial function caused by lipotoxic accumulation is linked to NAFLD and insulin resistance [42,43]. When the glucose uptake is decreased in the muscle, the liver stimulates de novo lipogenesis from glucose which leads to enhanced insulin resistance. In the milieu of insulin resistance, the lipolysis in adipose tissue is not suppressed which contributes to an increase in circulating free fatty acids [44,45].

It is well known that insulin resistance has a pivotal role in PCOS [46,47]. While there is limited knowledge about the molecular mechanisms underlying insulin resistance in PCOS, various studies have demonstrated how insulin signalling is impaired in the skeletal muscles, which are responsible for the insulin-resistance phenotype in PCOS women. Pioneered studies revealed abnormalities in the proximal part of the insulin-signaling pathway [48,49] and/or glycogen storage [50]. These studies include overweight or obese women on whom it is very difficult to differentiate whether insulin resistant occurs as a consequence of being overweight or having PCOS. The study which includes lean PCOS women revealed that the proximal part of insulin signaling may not be disturbed [51]. Hansen et al. [51] settled a case-control study to underline the mechanism of insulin resistance in lean women with PCOS. They performed hyperinsulinemic-euglycemic clamp and skeletal muscle biopsies with expression and phosphorylation analysis of protein included in the insulin-signaling cascade. They obtained that although there is lipid accumulation in skeletal muscles, insulin resistance in lean PCOS women with hyperandrogenism is not a consequence of injuries in the proximal part of the insulin-signaling cascade in skeletal muscles. In these women, insulin resistance is modulated by plasma adiponectin levels which influence skeletal muscles via AMPK. It has also been observed that absent phosphorylation of pyruvate dehydrogenase may contribute to reduced whole-body insulin resistance in PCOS women.

Insulin resistance is present in about 50–80% of women with PCOS and NAFLD [52]. The PCOS women with NAFLD have a higher prevalence of insulin resistance compared to PCOS women without NAFLD [53]. Also, insulin resistance is associated with a more severe form of liver steatosis and elevated alanine aminotransferase, a marker of liver injury well correlated with NAFLD [54]. 

It has been reported that obesity and adipose tissue dysfunction are associated with NAFLD and PCOS [36,47]. Tantanavipas et al. [55] evaluated the predictive factors of NAFLD in PCOS women. They observed that NAFLD is only predicted by waist circumference greater than 80 cm and that abdominal obesity, rather than the presence of PCOS, is independently associated with NAFLD. Vassilatou et al. [56] investigated performances of visceral adiposity index as a diagnostic marker for NAFLD in premenopausal PCOS women. They also compared its diagnostic performance with fatty liver index, lipid accumulation product, and hepatic steatosis in premenopausal PCOS women with NAFLD. The cross-sectional, case-control study included 145 premenopausal women with PCOS and 145 healthy control women matched by body mass index and age range. The fatty liver index, lipid accumulation product, and hepatic steatosis index were higher in women with NAFLD. Although the visceral adiposity index had a lower performance for diagnosis of NAFLD compared to others, its diagnosis performance is similar to another diagnostic tool.

Women with PCOS have abdominal fat accumulation when compared with BMI-matched controls [57], but Villa et al. [58] concluded that the abdominal fat accumulation is not significant for development of PCOS, but rather adipose tissue dysfunction has a central role in metabolic dysregulations observed in PCOS women.

Baranova et al. [36] analyzed the relationship between a biomarker of apoptosis and adipokines in serum as well as RNA profiles in visceral adipose tissue of NAFLD patients with PCOS and compared it to patients who have NAFLD only. The women with NAFLD and PCOS have higher serum levels of M30 than only NAFLD women. This could be explained by an androgen-dependent proapoptotic milieu in PCOS which induces the progression of NAFLD in PCOS women. The expression of LDLR mRNA, which is influenced by hyperandrogenism, in adipose tissue is lower in PCOS women with NAFLD compared with NAFLD women. These findings give us a deeper insight into the role of adipose tissue dysfunction and proapoptotic milieu in PCOS women with NAFLD and the pathogenesis of both diseases.

Hyperandrogenaemia also has a role in the development of NAFLD in PCOS women. Hyperandrogenaemia is linked to insulin resistance and obesity [59,60], but it is still not clear in the mechanism of action in the development of NAFLD in PCOS women whether androgen has a direct influence on the development of NAFLD in PCOS or via insulin resistance. Deregulation of insulin signaling in ovaries in PCOS women leads to an increase in the secretion of androgens [61]. Insulin resistance also leads to a decrease of sex binding hormone synthesis and an increase in free androgens [32]. The androgens lead to a decrease in LDR receptor gene expression and therefore hepatic steatosis [62]. Apoptosis of hepatocytes is also caused by androgens and contributes to the progression of NAFLD [36]. It has been assumed that NAFLD could be developed in PCOS women with hyperandrogenemia even though obesity is absent [63]. Cai et al. [62] revealed that a high free androgen index is associated with NAFLD and liver fat content in PCOS women regardless of the presence of obesity and/or insulin resistance status. Kim et al. [63] settled a case-control study with 275 non-obese PCOS women and a control group of 892 non-obese women aimed at comparing the prevalence of NAFLD in PCOS and non-PCOS women and determining the independent association between NAFLD and PCOS. They obtained that NAFLD was associated with PCOS and levels of androgenicity. Metabolic dysregulation is not only associated with fat content and body weight in PCOS women but is also modulated by androgens excess [64].

On the other hand, this standpoint become controversial after results reported by Macut et al. [32] that there is no significant difference in the prevalence of NAFLD in the hyperandrogenaemic and nonhyperandrogenaemic PCOS women.

Other mechanisms involved in the pathogenesis of NAFLD in PCOS women include inflammation [65,66,67,68], aberrant secretion of adipocytokines [69], and the influence of genetic polymorphisms [70] on susceptibility for NAFLD development in PCOS women. Inflammation is linked to the pathogenesis of PCOS [65]. It has been assumed that low-grade inflammation mediates insulin resistance in PCOS women [65,66]. Also, inflammation has a role in the maintenance of NAFLD and the progression of this benign condition to liver cirrhosis and hepatocellular carcinoma [67,68]. The genetic polymorphisms linked to NAFLD in PCOS could be responsible for observed inter-ethnic differences in the prevalence of NAFLD in PCOS. Table 2 presents known pathophysiological mechanisms involved in the development of NAFLD in PCOS women.

## 5. Diagnosis and Management of NAFLD in PCOS and Future Investigations

The diagnosis of NAFLD in women with PCOS leads to the prevention of further progression of the disease and reduces the risk for the development of pathologies linked to NAFLD.

There are numerous diagnostic approaches for NAFLD. However, the liver biopsy is the gold standard for diagnosis of NAFLD, although there are concerns about variability in sampling technique, invasiveness of the procedure itself, and relatively high costs. Considering all of the forementioned concerns regarding a liver biopsy, there are serum biomarkers and non-invasive imaging techniques for the diagnosis of steatosis, NASH, or advanced fibrosis. Ultrasonography is a first-line screening tool for the diagnosis of steatosis in the population at risk. The diagnosis of NAFLD is made by exclusion of other chronic liver diseases and causes of steatosis. A NAFLD-Fibrosis Score or transient elastography are used to identify advanced fibrosis. Although magnetic resonance can accurately measure status and fibrosis stage, it is not used in routine practice [71]. Future investigations must provide a novel invasive diagnostic approach for diagnosis of NASH and NAFLD and for the prediction of inflammation. It could lead to the development of a new class of drugs and personalization of NAFLD treatment.

Unfortunately, there is still no approved pharmacological treatment for NAFLD. The existing available treatment is change of lifestyle behaviors. The understanding of molecular mechanisms involved in the pathogenesis of NAFLD in PCOS leads to the development of new classes of drugs [72]. The dominant way is targeting the fat storage in the liver, and the medications in this group include peroxisome proliferator-activator receptor agonists (e.g., pioglitazone, elafibranor, saroglitazar), medications targeting the bile acid-farnesoid X receptor axis (obeticholic acid), inhibitors of fibroblast growth factor (FGF)-21 or FGF-19 analogs, de novo lipogenesis (aramchol, NDI-010976), and glucagon-like peptide (GLP-1) agonist (liraglutide) [72].

Rakoski et al. [73] performed a meta-analysis to evaluate the effect of pioglitazone on NAFLD. They observed that pioglitazone decreases liver steatosis but that they have no effect on liver fibrosis and inflammation. It has been observed that treatment with a GLP-1 receptor antagonist improves body mass index, adiponectin, and liver fat fraction in NAFLD patients [74]. The second group of drugs targets inflammation and oxidative stress. This group includes antioxidants (vitamin E), medications with a target in the tumor necrosis factor α (TNF α) pathway (emricasan, pentoxifylline), and immune modulators (amlexanox, cenicriviroc) [72]. Vitamin E decreases steatosis and inflammation, but it has no influence on liver fibrosis in patients with NAFLD [75]. There is no study about the treatment of NAFLD in PCOS women with vitamin E. The third group includes antiobesity agents such as orlistat or gut microbiome modulators (IMM-124e, fecal microbial transplant, solithromycin). According to results from a meta-analysis reported by Wang et al. [76], orlistat improves biochemical markers of liver damage and could be the drug of choice for the palliative treatment of NAFLD but not as a first-choice treatment. Antifibrotics (simtuzumab and GR-MD-02) could be considered as the fourth target group for NAFLD treatment.

Further investigation on the pharmacological treatment of NAFLD, especially NAFLD in PCOS, is needed considering the public health significance of both the disease and prevalence worldwide. Possible treatments for NAFLD in PCOS will not consist of just one drug or one class drug, but most likely it will be a combination of drugs that target different mechanisms involved in the pathogenesis of NAFLD and PCOS or both. The drugs used for NAFLD management are presented in Table 3.

## 6. The Potential Molecular Link between NAFLD and PCOS

The recent findings refer to the role of mitochondria and mitochondrial dysfunction in PCOS development. It is considered that mitochondrial dysfunction leads to oxidative stress, which is the driver of disturbance in insulin resistance, lipid metabolism, and follicle development. Also, abnormal mitochondrial DNA copy numbers and mutations in the mitochondrial gene are seen in women with PCOS [77]. Cree Green reported mitochondrial dysfunction in PCOS women with normal BMI [78]. One of the considered mechanisms of NAFLD development is the oxidation of biomolecules by mitochondrial reactive oxygen species which lead to increased oxidative damage of the liver [79]. The role of mitochondrial dysfunction and the investigation of pathological mechanisms which link PCOS and NAFLD are needed.

The gut microbiome is linked to insulin resistance which has a pivotal role in PCOS development [80]. Lindheim et al. [81] reported lower diversity and disturbance in the phylogenetic profiles in stool microbiome of PCOS women. These unfavorable changes in the microbiome are linked to reproductive and metabolic defects in women with PCOS [81].

The liver is exposed to the intestinal bacteria because the liver accepts blood from the gut. It is assumed that endotoxins produced by Gram-negative bacilli may be involved in NAFLD pathogenesis. It is suggested that intestinal (impaired intestinal barrier function and dysbiosis with increased abundance of ethanol-producing bacteria) and hepatic factors (hyperleptinemia which leads to the excessive response to endotoxins, intrahepatic inflammation, and fibrosis) may contribute to NAFLD development [82].

It is known that besides a genetic predisposition and diet, the gut microbiome influences liver carbohydrate and lipid metabolism and the balance between pro-inflammatory and anti-inflammatory mediators in the liver [83].

Recent studies revealed that overactivation of the endocannabinoid system is linked to insulin resistance and the progression of PCOS [84,85]. It has also been reported that genetic polymorphism of cannabinoids receptor 1 influences the hyperandrogenemia state in PCOS women and the development of NAFLD [86,87]. Antagonism of cannabinoids receptor 1 leads to the improvement of hepatic steatosis and insulin resistance [88].

Table 4 presents possible new pathophysiological mechanisms involved in the development of NAFLD in women with PCOS.

Figure 1 presents established and possible new pathophysiological mechanisms involved in PCOS and NAFLD development.

Reported findings on the role of mitochondrial dysfunction, gut microbiome, and the endocannabinoid system are possible key targets for further investigations about pathophysiology as well as diagnosis and pharmacotherapy of NAFLD in PCOS.

## 7. Conclusions

The mechanism of development of NAFLD in PCOS women is not fully known. Besides the progress in the diagnosis of NAFLD in PCOS, there is a lack of knowledge about mechanisms that lead to the development of NAFLD in PCOS. Future studies which would integrate epidemiological, clinical, and molecular investigations about NAFLD in PCOS will have a key role in the development of new diagnostic and therapeutic approaches of NAFLD in PCOS.

## Figures and Tables

**Figure 1 biomedicines-10-00131-f001:**
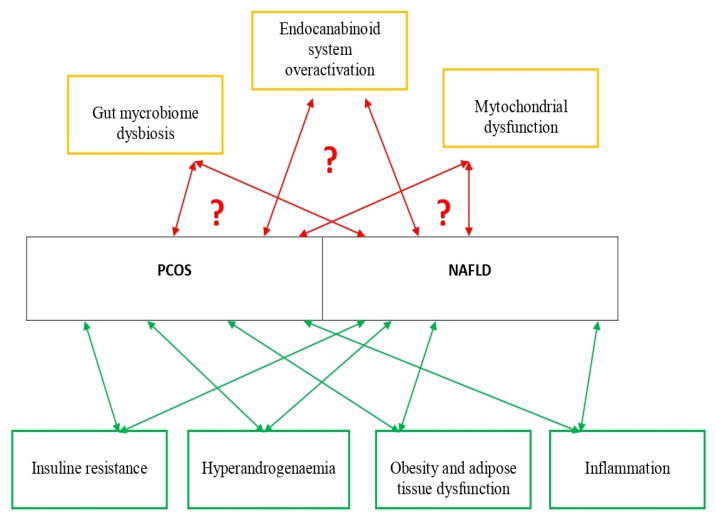
Established and possible new pathophysiological mechanisms involved in PCOS and NAFLD development. Studies about established pathophysiological mechanisms involved in PCOS and NAFLD are presented in Table 1. Studies about possible new pathophysiological mechanisms involved in PCOS and NAFLD are presented in Table 2.

**Table 1 biomedicines-10-00131-t001:** Epidemiology and risk factors for development of NAFLD in PCOS women.

Studies	Epidemiology	Risk Factors
**Won et al.** [24], retrospective cohort study	Prevalence of NAFLD in study population (586 women diagnosed with PCOS) was 8.7% (51/586).	MetS diagnosis (hazard ratio [HR] 5.6, 95% confidence interval [CI] 2.2–14.4, *p* < 0.01) Hyperandrogenism (HA) (HR 4.4, 95% CI 1.4–13.4, *p* = 0.01)
**Romanowski et al.** [25], case-control study	NAFLD was present in 23.8% of the PCOS group (101). At control group (33), it represented 3.3%, (*p* = 0.01). PCOS group (101 women) was subdivided into two subgroups: PCOS+NAFLD (24) and PCOS (77)	BMI, waist circumference, glucose intolerance, insulin levels was higher in PCOS+NAFLD group compared to only PCOS group.
**Asfari et al.** [26], National Inpatient Sample database between 2002 and 2014	77,415 of 50,785,354 women (0.15%) had PCOS.	Patients with PCOS had significantly higher rates of NAFLD (OR 4.30, 95% CI 4.11 to 4.50, *p* < 0.001).
**Shengir et al.** [27], cross-sectional cohort study, 101 women with diagnosed PCOS	Prevalence of NAFLD and liver fibrosis was 39.6% and 6.9%, respectively, in the study population.	Higher body mass index (adjusted odds ratio (aOR) 1.30, 95% CI: 1.13–1.52). Hyperandrogenism (aOR: 5.32, 95% CI: 1.56–18.17). Elevated ALT (aOR: 3.54, 95%CI: 1.10–11.47).
**Salva-Pastor et al.** [28], cross-sectional study, with 98 women with diagnosed PCOS (Rotterdam 2003 criteria), Controls were matched by age and body mass index (BMI)	Prevalence of NAFLD was markedly higher in patients with than without PCOS at 69.3% vs. 34.6%, respectively. NAFLD prevalence was 84.3% in PCOS patients with phenotype A, while in another phenotype, it was 41.1%.	Hyperandrogenism (OR 21.8) and BMI (OR 11.7) are risk factors for developing NAFLD in patients with PCOS.
**Vassilatou et al.** [29], Prospective, observational, and cross-sectional study	NAFLD was detected in 71/110 women (64.5%). Women with NAFLD compared to women without NAFLD were more commonly diagnosed with PCOS (43.7% vs. 23.1%, respectively), metabolic syndrome (30.2% vs. 5.3%), and abnormal lipid profile (81.1% vs. 51.3%).	HOMA-IR values (OR 2.2, 95% CI: 1.1–4.4) and triglyceride levels (OR 1.01, 95% CI: 1.00–1.02) are independent predictor factors for NAFLD
**HarshaVarma et al.** [30], prospective, cross-sectional study 60 women with PCOS (Rotterdam 2003 criteria)	23 (38.3%) women with PCOS had NAFLD.	HOMA IR Hyperandrogenemia
**Sarkar et al.** [31], Retrospective study of 102 women with biopsy-confirmed NAFLD between 2008–2019	36% (37 women) of study group had PCOS.	PCOS was risk factor for severe hepatocyte ballooning (OR 3.4, 95% CI 1.1–10.6, *p* = 0.03) and advanced fibrosis (OR 7.1, 95% CI 1.3–39, *p* = 0.02).
**Macut et al.** [32], cross-sectional study included 600 Caucasian women diagnosed with PCOS (Rotterdam criteria)	NAFLD was more prevalent in patients with PCOS than in controls (50.6% vs. 34.0%, respectively).	HOMA-IR and lipid accumulation products were independently associated with NAFLD (*p* ≤ 0.001).
**Rocha et al.** [4], Meta-analysis of 17 studies published between 2007 and 2017 that included 2734 PCOS patients and 2561 controls of similar age and body mass index (BMI)	PCOS patients have increased prevalence of NAFLD (OR 2.54, 95% CI 2.19–2.95). PCOS women with hyperandrogenism (classic phenotype) have a higher prevalence of NAFLD compared to women with PCOS without hyperandrogenism, even after correction for confounding variables.	Hyperndrogenism
**Shengir et al.** [33], Meta-analysis of 23 studies with 7148 participants	South American/Middle East PCOS women had a greater risk of NAFLD than women of European and Asia origin.	PCOS women had a 2.5-fold increase in the risk of NAFLD compared to controls (pooled OR 2.49, 95% CI 2.20–2.82). BMI seems to be the main cofactor.
**Wu et al.** [34], Meta-analysis of 17 studies	PCOS is significantly associated with high risk of NAFLD.	PCOS patients with hyperandrogenism had a significantly higher risk of NAFLD compared with controls (OR 3.31, 95% CI = 2.58–4.24).

**Table 2 biomedicines-10-00131-t002:** Pathophysiological mechanisms involved in the development of NAFLD in PCOS women.

Factors Influencing the Development of PCOS and NAFLD	Studies
Insulin resistance	Petta et al. [35], Kotronen et al. [37], Zeng et al. [47], Højlund et al. [48], Corbould et al. [49], Glintborg et al. [50], Hansen et al. [51], Baranova et al. [52]
Obesity and adipose tissue dysfunction	Baranova et al. [36], Tantanavipas et al. [55], Vassilatou et al. [56], Zhu et al. [57], Villa et al. [58]
Hyperandrogenaemia	Kim et al. [63], Cai et al. [62], Condorelli et al. [64]
Inflammation	Shorakae et al. [65], Regidor [66], Luci [67], Møhlenberg [68]

**Table 3 biomedicines-10-00131-t003:** The groups of drugs used in NAFLD management.

Groups of Drugs for NAFLD Menagment	Papers about Drugs for NAFLD Menagment
Peroxisome proliferator-activator receptor agonists (e.g., pioglitazone, elafibranor, saroglitazar) Medications targeting the bile acid-farnesoid X receptor axis (obeticholic acid) Inhibitors of fibroblast growth factor (FGF)-21 or FGF-19 analogs and de novo lipogenesis (aramchol, NDI-010976) Glucagon-like peptide (GLP-1) agonist (liraglutide)	Rotman et al. [72], Rakoski et al. [73], Fan et al. [74]
Antioxidants (vitamin E), medications with a target in the tumor necrosis factor α (TNF α) pathway (emricasan, pentoxifylline), and immune modulators (amlexanox, cenicriviroc)	Rotman et al. [72], Sanyal et al. [75]
Antiobesity agents such as orlistat or gut microbiome modulators (IMM-124e, fecal microbial transplant, solithromycin) Antifibrotic (simtuzumab and GR-MD-02)	Rotman et al. [72], Wang et al. [76]

**Table 4 biomedicines-10-00131-t004:** Possible new pathophysiological mechanisms involved in the development of NAFLD in PCOS women.

Possible Factors Influencing the Development of PCOS and NAFLD	Studies
Mitochondrial dysfunction	Zeng et al. [77], Cree-Green et al. [78], Simões et al. [79]
Gut microbiome dysbiosis	Jiao et al. [80], Lindheim et al. [81], Kessoku et al. [82], Kolodziejczyk et al. [83]
Endocanabinoid system overactivation	Juan et al. [84], Cui et al. [85], Kuliczkowska Plaksej et al. [86], Jędrzejuk et al. [87]

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
