# Peer review of "Prevalence, Risk Factors, and Pathophysiology of Nonalcoholic Fatty Liver Disease (NAFLD) in Women with Polycystic Ovary Syndrome (PCOS)"

_biomedicines, 2022, doi:10.3390/biomedicines10010131_

Round 1

Reviewer 1 Report

This paper describes the relation between PCO, Insulin resistance, inflammation and NAFLD.  This is done well however some small suggestions:

Please add this to the references and discuss more in details the leak in the gut, bakerial contamination in the hepatic blood and alcohol production in the gut. "Kessoku T, Kobayashi T, Tanaka K, Yamamoto A, Takahashi K, Iwaki M, Ozaki A, Kasai Y, Nogami A, Honda Y, Ogawa Y, Kato S, Imajo K, Higurashi T, Hosono K, Yoneda M, Usuda H, Wada K, Saito S, Nakajima A. The Role of Leaky Gut in Nonalcoholic Fatty Liver Disease: A Novel Therapeutic Target. Int J Mol Sci. 2021 Jul 29;22(15):8161. doi: 10.3390/ijms22158161. PMID: 34360923; PMCID: PMC8347478."

Further a shorter version of the case described would make it more clear

Reviewer 2 Report

The review is relevant for the field and well organised. The review a) covers the epidemiology, pathology, diagnosis and treatment of NAFLD in PCOS women comprehensively; b) the references and methodology applied to choose them are appropriate and nicely described, and c) the cited case reports are well contextualized. Since this review mainly discusses the commonalities/associations between AFLD and PCOS, I suggest to alter the title of this review to emphasize precisely that link between these diseases. Suggestions:

 “Prevalence, risk factors and pathophysiology of non-alcoholic fatty liver disease (NAFLD) in women with polycystic ovary syndrome (PCOS)”.

“Pathophysiology and molecular links between polycystic ovary syndrome (PCOS) and non-alcoholic fatty liver disease (NAFLD).”

Main comments:

1) Overall, the review would benefit from improved English and writing style. Several sentences throughout the whole manuscript - I would say a significant proportion of the text – are too short, and it would make it a smoother reading if these sentences are linked up to a preceding or succeeding sentence, by a comma, semi-colon and/or linkup words such as ‘in which’, ‘where’, ‘as’, ‘since’, ‘given that’, ‘because’, etc. There are too many examples and I won’t list them here, but advise the authors to revise the manuscript accordingly.

2) I suggest to include a table summarising information on the following sections: 3. Epidemiology and predictors of NAFLD in PCOS women, 4. Pathophysiology of NAFLD in PCOS, and 5. Diagnosis and management of NAFLD in PCOS and sight in future. I mean a table similar to the Table 1, but which would include actual values (or a range of values) for the different prevalence and risk factors covered. For example, the table would show prevalence of NADLF in PCOS women as “a to b %”, and prevalence of NAFLD in PCOS-free women as “a-x to b-x %”. It seems reasonable to do this also for quantitative information on the risk factors, and across sections 3-5, to make it easier to retain some useful information about the topic, and to avoid redundancy. For instance, the sentence “PCOS women with NAFLD have a higher prevalence of insulin resistance compared to PCOS women without NAFLD”, and variations of this sentence, are mentioned several times in the manuscript.

3) I suggest to modify Figure 1. Please include ‘established’ mechanisms underlying PCOS and NAFLD development, in addition new possible mechanisms shown in Figure 1, and colour-code the new/known mechanisms, citing references on the known mechanisms.

4) Section 7 (7. The case of a young patient with PCOS and NAFLD) seems out of place in the manuscript. Please, justify the need to include this section, and consider including it elsewhere.

Minor comments:

Below, I list some suggestions/corrections on the English/writing style.

Lines 85-86: “The patient underwent to diet and exercise program which lead to weight loss and normalization of her aminotransferase level”, correct to: normalization of aminotransferase levels.

Lines 88-89: “numerous studies investigate the prevalence”, correct to: numerous studies have investigated the prevalence.”

Lines 92-93: “The study included 586 women with PCOS. PCOS phenotype, metabolic syndrome…were documented”, correct to: The study included 586 women with PCOS, which documented the PCOS phenotype, metabolic syndrome…in the cohort”.

Lines 95-97: “They have observed that MetS diagnosis and hyperandrogenism were the risk factor associated with NAFLD occurrence, but also insulin level in 75 g glucose tolerance test well as body mass index was not. Correct to: They have observed that Met diagnosis…., whereas insulin levels (assessed through the 75g glucose tolerance test) and body mass index were not significant risk factors.”

Lines 105-107: “PCOS women with NAFLD have higher age, weight, BMI, abdominal circumference, insulin levels, and glucose tolerance test results than PCOS women without NAFLD”. Correct to: PCOS women with NAFLD have higher age…, and glucose intolerance than PCOS women without NAFLD.

Line 110: “The conducted study include 50 785 354 women”. Correct to: The study included 50 785 354 women.

Line 114: “from South Asia, population with severe NAFLD and PCOS.” Correct to: from South Asia, a population with severe NAFLD and PCOS.

Lines 119-122: “After multiple regression and adjusting for duration of PCOS and insulin resistance measured by homeostasis model for assessment of insulin resistance, the independent predictors for development of NAFLD in South Asian women with PCOS were independent predictors of NAFLD were higher body mass index, hyperandrogenism and elevated ALT.” This sentence does not make sense with the underlined words in it.

Lines 132-133: “The prevalence of NAFLD was significantly higher in phenotype A (about 84.3%) while the prevalence of NAFLD in other phenotypes was about 41.1%.”. Clarify what phenotype A is.

Lines 133-134: “The authors suggested NAFLD screening for all PCOS patients, except in PCOS women with BMI < 25 and without hyperandrogenism.” It seems irrelevant to mention this information from the cited study.

Line 142: “Varma et al. [30] found the prevalence of NAFLD about 38.3% in PCOS women”. Correct to: Varma et al. [30] found that the prevalence of NAFLD is about 38.3% in PCOS women.

Line 143: “They reported that serum total cholesterol. Insulin, HOMA-IR”. Replace the underlined ‘full stop’ by a comma.

Lines 145-146: “Sarkar et al. [31] reported that PCOS is associated with severe NASH, including fibrosis. According to that, PCOS women should be screened for NAFLD.” Please, spell out the acronym ‘NASH’, and alter the sentence to: Sarkar et al. [31] reported that PCOS is associated with severe NASH, including fibrosis, suggesting that PCOS women should be screened for NAFLD.

Line 148: “Macut et al. [31]evaluated the factor associated with NAFLD in Caucasian women” Correct to: Macut et al. [31]evaluated the risk factors associated with NAFLD in Caucasian women.

Line 172: “We et al. [34] settled the meta-analysis to research is the association between NAFLD and PCOS consequence of shared risk factors or PCOS in an independent factor for NAFLD development.” This sentence does not make sense.

Lines 192-194: “When uptake of glucose by muscle decrease, the liver start de novo syntheses of lipids from glucose what lead to enhances of insulin re-193 sistance.” Correction: When the glucose uptake is decreased in the muscle, the liver stimulates de novo lipogenesis from glucose which leads to enhanced insulin resistance.

Linea 196-198: “There is a lack of knowledge in molecular mechanisms that contribute to the development of insulin resistance in PCOS, but generally, the state is that impairs insulin signaling in skeletal muscle is responsible for insulin resistance in PCOS women.” Suggestion: While there is limited knowledge on the molecular mechanisms underlying insulin resistance in PCOS, various studies have demonstrated how insulin signalling is impaired in the skeletal muscle, which is responsible for the insulin resistance phenotype in PCOS women.

Lines 201-203: “These studies include overweight or obese women in whom is very difficult to differentiate is insulin signaling impairment consequences of overweight or PCOS.” Correct to : These studies include overweight or obese women on whom it is very difficult to differentiate whether insulin resistant occurs as a consequence of overweight or PCOS.

Round 2

Reviewer 2 Report

The authors have addressed all my comments. The new Tables and Figure 1 are appropriate; well done.

Regarding Section 7 and in response to the authors’ comments:

The ‘young patient’ case would fit better elsewhere (other than Introduction), such as embedded in any of the 6 sections presented. I would do that, or remove the ‘young patient’ case from the manuscript, which I leave to the authors’ discretion.

Minor corrections:

Table 2: “Obeity and adipose tissue dysfunction”. Correct Obeity to Obesity.

Section 8 (Conclusion) should now read section 7.